# Natural Extracts to Augment Energy Expenditure as a Complementary Approach to Tackle Obesity and Associated Metabolic Alterations

**DOI:** 10.3390/biom11030412

**Published:** 2021-03-10

**Authors:** Marina Reguero, Marta Gómez de Cedrón, Guillermo Reglero, José Carlos Quintela, Ana Ramírez de Molina

**Affiliations:** 1Molecular Oncology Group, Precision Nutrition and Health, IMDEA Food Institute, CEI UAM + CSIC, Ctra. de Cantoblanco 8, 28049 Madrid, Spain; marina.reguero@imdea.org; 2NATAC BIOTECH, Electronica 7, 28923 Madrid, Spain; jcquintela@idoasis.es; 3Production and Characterization of Novel Foods Department, Institute of Food Science Research CIAL, CEI UAM + CSIC, 28049 Madrid, Spain; guillermo.reglero@imdea.org

**Keywords:** precision nutrition, metabolic stress, inflammation, natural extracts, energy expenditure, thermogenesis, chronic diseases

## Abstract

Obesity is the epidemic of the 21st century. In developing countries, the prevalence of obesity continues to rise, and obesity is occurring at younger ages. Obesity and associated metabolic stress disrupt the whole-body physiology. Adipocytes are critical components of the systemic metabolic control, functioning as an endocrine organ. The enlarged adipocytes during obesity recruit macrophages promoting chronic inflammation and insulin resistance. Together with the genetic susceptibility (single nucleotide polymorphisms, SNP) and metabolic alterations at the molecular level, it has been highlighted that key modifiable risk factors, such as those related to lifestyle, contribute to the development of obesity. In this scenario, urgent therapeutic options are needed, including not only pharmacotherapy but also nutrients, bioactive compounds, and natural extracts to reverse the metabolic alterations associated with obesity. Herein, we first summarize the main targetable processes to tackle obesity, including activation of thermogenesis in brown adipose tissue (BAT) and in white adipose tissue (WAT-browning), and the promotion of energy expenditure and/or fatty acid oxidation (FAO) in muscles. Then, we perform a screening of 20 natural extracts (EFSA approved) to determine their potential in the activation of FAO and/or thermogenesis, as well as the increase in respiratory capacity. By means of innovative technologies, such as the study of their effects on cell bioenergetics (Seahorse bioanalyzer), we end up with the selection of four extracts with potential application to ameliorate the deleterious effects of obesity and the chronic associated inflammation.

## 1. Introduction

The World Health Organization (WHO) has alerted the world about the increase of chronic diseases, such as obesity, diabetes, cardiovascular diseases, cancer, and chronic respiratory diseases, which have been linked as the main causes of death in the 21st century (WHO 2019 https://apps.who.int/iris/handle/10665/311696). Nowadays, obesity is an epidemic of western countries that has complex and multifactorial causes, some being associated with lifestyle and imbalanced diets, but others with the genetic background of each person [1].

Obesity, and more specifically visceral obesity, leads to the development of numerous diseases such as type 2 diabetes, insulin resistance, liver steatosis, cardiovascular disease, and even cancer. In the course of obesity, atrophy in adipocytes promotes massive production of pro-inflammatory mediators, which recruit inflammatory macrophages, leading to the ectopic fatty acid accumulation in other metabolic organs such as liver, pancreas, and muscles [2,3,4]. In addition, obesity aggravates the infection of some pathogens, as is, for instance, the case of SARS-CoV-2, associated with the worst pandemic regarding humanity in the current century [5].

Regular practice of physical exercise and compliance with healthy nutritional patterns are included in all recommendations for the maintenance of optimal health conditions [6,7,8]. However, it has been proven that exercise and healthy nutrient recommendations do not affect individuals in the same way, as these recommendations do not take into consideration the genetic heterogenicity between individuals [9,10,11], such as genetic variants (single nucleotide polymorphisms, SNPs), nor the nutritional and metabolic status of individuals, such as obesity, dyslipidemia, insulin resistance metabolic syndrome, cardiovascular disease, and cancer. All these factors will shape the final metabolic effects of diet-derived ingredients. Precision Nutrition integrates by one hand the knowledge of the diversity in the genomes, which influences nutrient bioavailability and their metabolism [1,12,13,14], and by the other hand, the knowledge of how nutrients may affect the expression of genes in critical metabolic pathways. In addition, Precision Nutrition takes into account factors related to lifestyle such as diet, exercise, alcohol consumption, as well as the nutritional and metabolic status of the individuals [15,16]. In this regard, for example, several SNPs have been associated with obesity and muscle performance [17,18], and nutritional strategies based on the knowledge of the effect of bioactive compounds from natural sources such as plants, at the molecular level may contribute to restore the metabolic balance within individuals susceptible to develop chronic diseases [19,20].

Plant-derived bioactive compounds have been extensively studied for their antioxidant effect [21]. For instance, γ-oryzanol from rice has been proved to reduce the low grade of inflammation that comes with LPS-induced cognitive and memory impairments [22]. Another example widely studied is resveratrol, which has been shown to reduce obesity inflammation in skeletal muscle [23]. These anti-inflammatory and antioxidant capacities are crucial to control the levels of reactive oxygen species (ROS) and inflammation generated during obesity, T2DM, cardiovascular diseases, dyslipidemias, and cancer [24,25,26]. Although the identification of the bioactive compounds responsible for the beneficial effects is important, in the last years, it has been shown that natural bioactive extracts are potent mixtures of bioactive compounds which may act synergistically by targeting different molecular pathways. Therefore, the study of the capacity of natural extracts to control inflammatory signaling pathways, nutrient catabolism, or cellular oxidative capacity, is necessary for their successful application against chronic metabolic disorders.

In this sense, obesity, and the associated low grade of chronic inflammation could be a good target for the study of natural extracts. The discovery of the existence of metabolically active brown adipocyte tissue (BAT) in humans [27], opens novel possibilities to control obesity-related metabolic alterations, and therefore, to reduce the low grade of chronic inflammation at the systemic level, by mean of the enhancement of thermogenesis. In addition, browning of the white adipose tissue (WAT), with BAT characteristics, may increase the general thermogenic capacity within the body [28,29]. The upregulation of Sirtuin 1 (SIRT1)/Protein Kinase AMP-Activated Catalytic (AMPK) pathways, and overexpression of Peroxisome Proliferator Activated Receptor Gamma coactivator 1 Alpha (PGC1a) and Uncoupling Protein 1 (UCP1), have been demonstrated to promote thermogenesis in BAT as well as to increase the browning of WAT, which contributes to reduce fatty acids accumulation in preclinical mouse models [30].

On the other hand, muscles also play a key role in the overall energy expenditure, as it is the major energy consumer within the body, during physical exercise. Interestingly, muscles also have specific calcium-dependent thermogenesis [31,32].

Thus, as there are no successful remedies for obesity except surgeries or drugs with many side effects, the understanding of chronic inflammation associated with obesity and the regulation of thermogenesis at the molecular level, will allow the development of nutritional intervention strategies based on the application of nutritional genomics. Herein, we aim to provide molecular targets of bioactive natural extracts, EFSA approved, to ameliorate the low grade of chronic inflammation and metabolic and oxidative stress associated with obesity. Therefore, we perform the screening of 20 natural extracts derived from different plants to activate thermogenesis in BAT and/or browning of WAT. Additionally, we also investigate their effects on the enhancement of mitochondrial respiration in muscles as a complementary approach to augment the energy expenditure at the systemic level (Figure 1). For the achievement of this goal, and in the frame of Precision Nutrition, we first select genes with well-known SNPs associated with the increased susceptibility to develop metabolic alterations (*nutrigenetics*). Then, we investigate the potential of natural extracts to modulate their expression (*nutrigenomics*).

As indicated previously, we pay special attention to natural extracts with the ability to augment thermogenesis and energy expenditure to alleviate the inflammation and metabolic stress associated with obesity. Through the analysis of cell bioenergetics, we functionally demonstrate their effect on the activation of thermogenesis, by means of the quantification of the H^+^ leak levels, and the enhanced oxidative phosphorylation at mitochondria, in two in vitro models: human differentiated adipocytes and myocytes.

## 2. Materials and Methods

### 2.1. Cell Lines and Reagents

Human Skeletal Muscle Myoblast cell line (HSMM), Skeletal Muscle Cell Growth Basal Medium (SKBM) were purchased from Lonza Iberica (Madrid, Spain). The human Simpson-Golabi-Behmel syndrome (SGBS) pre-adipocyte cell strain was obtained from the ATCC.

Dulbecco’s Modified Eagle’s Medium (DMEM) and Dulbecco’s Modified Eagle’s Medium/Nutrient F-12 Ham (DMEM-F12) were purchased from Gibco. Penicillin/streptomycin, L-glutamine, Biotin, pantothenic acid, rosiglitazone, human apo-transferrin, human insulin, dexamethasone, 3-Isobutyl-1-methylxanthine (IBMX), cortisol, 3,3′,5-Triiodo-L-thyronine (T3), 3-(4,5-dimethylthiazol-2-yl)-2,5-diphenyltetrazolium bromide (MTT), Carbonyl cyanide-4-(trifluoromethoxy)phenylhydrazone (FCCP), rotenone, antimycin A, forskolin, paraformaldehyde, and qiazol were purchased from Sigma–Aldrich (Merck, Madrid, Spain).

Fetal bovine serum (FBS) and horse serum (HS) were purchased from Gibco (Thermofisher Scientific, Madrid, Spain).

XFe DMEM (pH7.4) assay medium, and 96-well XF96 cell culture microplates and cartridges were purchased from Seahorse Bioscience-Agilent Technologies (Madrid, Spain).

TaqMan reverse transcription reagents kit, Taqman gene expression master mix, and Sybr green were purchased from Thermofisher Scientific (Applied Biosystems, Madrid, Spain). Primers were designed using NCBI tools and were purchased from Isogen. TaqMan Gene Expression Array Cards were designed and purchased from Thermofisher Scientific (Madrid, Spain).

### 2.2. Natural Plant-Derived Extracts

A list of 20 EFSA approved plant-derived extracts for human consumption was selected from the library of Natac Biotech, mainly focusing on the bioactive activity of the main molecule, but always taking into account the importance of the complete extract.

Table 1 displays the list of these 20 natural extracts selected in this study, indicating their natural source, the part of the plant where they were obtained, their main bioactive compounds, and the extraction solvents used.

### 2.3. Cell Culture

#### 2.3.1. Cell Culture of Human HEK293

Human embryonic kidney cells 293 (HEK293) were maintained in DMEM containing 10% FBS and 100 U/mL penicillin/streptomycin at 37 °C in an incubator with 5% CO_2_ and 95% humidity. Cells were seeded in 24-well plates at a confluence of 50 × 10^4^ cells/well for RNA extraction. Cells were treated for 48 h with the extracts at the optimized concentrations, based on the previously determined IC_50_ (concentration which reduces 50% of cell viability) in triplicates.

#### 2.3.2. Human Adipocytes Cell Culture and In Vitro Differentiation

Human SGBS pre-adipocytes were seeded in 6-well plates (3 × 10^4^ cells/well) and cultured in advanced DMEM-F12 medium containing 33 μM biotin, 17 μM pantothenic acid, 100 U/mL penicillin/streptomycin, 1%L-glutamine, 2%HEPES buffer and 10%FBS at 37 °C in 5% CO_2_ and 95% humidity. Cells were kept for 6 days until they reached complete confluence, refreshing the media every two days. Then, differentiation was induced for four days using induction media composed by serum-free medium supplemented with 33 μM biotin, 17 μM pantothenic acid, 100 U/mL penicillin/streptomycin, 1% L-glutamine, 2% HEPES buffer, 2 μM rosiglitazone, 10 μg/mL human apo-transferrin, 20nM human insulin, 25 nM dexamethasone, 500 μM IBMX, 100 nM cortisol and 200 pM T3. On the fifth day, the medium was changed to the differentiation media, same as the induction media but without rosiglitazone, IBMX and dexamethasone. Cells were kept in the differentiation media for 10 days refreshing the media every three days [33,34]. At the terminal differentiation point, cells were treated with the extracts at 3 concentrations in triplicates, and the RNA extraction was performed 48 h after the treatment. CCAAT/enhancer-binding protein alpha (*CEBPA*) and peroxisome proliferator-activated receptor gamma (*PPARG*) were measured by qPCR as markers of adipocyte differentiation.

#### 2.3.3. Human Myocytes Cell Culture and In Vitro Differentiation

Human HSMM myoblasts were harvested in the SKGM2 commercial media at 37 °C in an incubator with 5% CO_2_ and 95% humidity. To induce myotube differentiation, 5 × 10^4^ HSMM cells were seeded 6-well plates in SKGM2 media until they reached confluence. Then, media was replaced into advanced DMEM-F12 and supplemented with 2% HS and 100 U/mL penicillin/streptomycin for 7 days, refreshing media each other day [35]. Differentiated myocytes were treated for 48 h with the different extracts at 3 concentrations in triplicates, previous to RNA extraction.

Myogenin (*MYOG*), myosin Heavy Chain 2 (*MYH2*), and myocyte enhancer factor-2 (*MEF2*) were used as markers of myocyte differentiation.

### 2.4. MTT Assay

HEK293T cells were cultured in 96-well plates at densities of 2000 cells/well overnight to allow the cells to attach. The next day cells were treated for 48 h with different plant extracts solved in DMSO diluted in their correspondent growth medium at different concentrations. Subsequently, the culture medium with extracts was removed and replaced by 200 μL of fresh growth medium containing 10% sterile filtered MTT at 5 mg/mL. After 3 h, the medium was removed, and the insoluble formazan crystals were dissolved in 100 μL/well of DMSO. Absorbance was measured at 560 nm in a Victor Nivo multimode plate reader, Perkin Elmer. The inhibition of growth (mitochondrial function) due to extracts was expressed as a percentage of viable cells in experimental wells relative to control wells, therefore calculating IC_50_ values. Three independent experiments were performed with each of the extracts.

### 2.5. Gene Expression Analysis

Total RNA was extracted with TriReagent (Sigma). One microgram of RNA was reverse-transcribed with the High Capacity RNA-to-cDNA Master Mix system (Life Technologies).

Quantitative polymerase chain reaction (qPCR) was performed in the 7900 HT Real-Time PCR System (Life Technologies) using the VeriQuest SYBR Green qPCR Master Mix or Taqman Gene Expression Master Mix, and oligos or Taqman probes were used. First, screening in HEK293T was performed in TaqMan gene expression customized array cards (TLDAs) with only one replicate per gene and condition. The validation of those results was performed in customized gene expression array cards in duplicates. For the rest of the assays, samples were used in triplicates. The 2^−DD^Ct method was applied to calculate the relative gene expression [36].

Glyceraldehyde-3-Phosphate Dehydrogenase (*GAPDH*) and/or Beta-2-Microglobulin (*B2M*) were used as endogenous controls. Appendix A summarize primers and Taqman probes used in the study.

### 2.6. Lipid Accumulation: Oil Red O Staining

Differentiated SGBS adipocytes were washed twice with PBS, fixed with 4% paraformaldehyde (PFA) at room temperature (RT) for 30 min, washed twice again with PBS and with 60% isopropanol. Then, cells were stained with 3 g/L oil red O (in 60% isopropanol) at RT for 30 min. After that, cells were washed three times with PBS and left to dry at RT for 10 min for visualizing using a microscope (Leica DM 2000 LED). Cells were resuspended in isopropanol, and absorbance was measured at 510 nm in a Victor Nivo multimode plate reader (Perkin Elmer).

### 2.7. Analysis of Mitochondrial Respiration

To analyze the mitochondrial respiration, we monitored the extracellular flux analysis of the Oxygen Consumption Rate (OCR) (Cell MitoStress Test) after the injection of several modulators of the electron transport chain, with the XFe96 Cell Bionalyzer (Seahorse Biosciences, XFe96). Optimal cell density and drugs titration were previously determined.

Prior to the experiments, cells were pre-treated with indicated doses of the extracts for 48 h. Non-treated cells were kept as controls.

For *MitoStress assay*, SGBS and HSMM cells were seeded onto 96-well XFe96 cell culture microplates at a density of 2500 and 4500 cells/well respectively. Cells were kept in their corresponding growth mediums until they reached confluence and then it was started the differentiation process. When terminal differentiation was reached, cells were treated for 48 h with different plant extracts solved in DMSO. Then, the medium was changed to 10 mM glucose for HSMM cells or 25 mM glucose for SGBS cells, and 2 mM glutamine and 1 mM pyruvate XFe DMEM (5 mM Hepes). Cells were incubated for 45 min at 37 °C without CO_2_. Three different modulators of mitochondrial respiration were sequentially injected. After basal oxygen consumption rate (OCR) determination (1–3 measurements), oligomycin (2.5 μM), which inhibits ATPase, was injected to determine the amount of oxygen dedicated to ATP production by mitochondria (3–6 measurements). To determine the maximal respiration rate or spare respiratory capacity, FCCP (carbonyl cyanide-4-(trifluoro-methoxy)phenyl-hydrazone) was injected (1.2 μM) to free the gradient of H+ from the mitochondrial intermembrane space (7–9 measurements) and thus to activate maximal respiration. Finally, antimycin A and rotenone (1 μM) were added to completely inhibit the mitochondrial respiration (10–12 measurements).

The analysis of mitochondrial oxidative phosphorylation was performed with 5–6 replicates per plate in 3–5 independent experiments.

### 2.8. Data and Statistical Analysis

One-way analysis of variance (ANOVA; Bonferroni post-hoc test) was used to determine qPCR differences in gene expression. * *p* < 0.05, ** *p* < 0.01, *** *p* < 0.005, and *** *p* < 0.001 indicate statistic significant differences. GraphPad Prim 8.0.1 statistical software was used for all statistical analyses. Results are expressed as mean ± standard deviation.

## 3. Results

### 3.1. Screening of Plant-Derived Extracts in HEK293T Cell Line: MTT Assay and Gene Expression of Selected Metabolic Pathway

A total of 20 extracts from natural sources, and EFSA approved for human consumption, were selected from a library from Natac Biotech. A first screening was performed in HEK293T, a non-transformed immortal cell line, to calculate IC_50_ values (concentration of the extracts able to diminish 50% of cell viability) as a readout of the bioactive potential of the extracts.

Therefore, as all the extracts had IC_50_ values above 100 ug/mL, we selected three concentrations, 10, 30, and 90 ug/mL below to that of the IC_50_, to be used in the analysis of their effects in the expression of genes implicated in metabolic processes of interest. In the case of carnosic acid from rosemary, hydroxytyrosol from olive, resveratrol, and proanthocyanins from grape, and curcuminoids from curcumin, as their IC_50_ values were much lower than 100 ug/mL, the working concentrations for the gene expression analysis were below 20 ug/mL. Hence, Table 1 shows the selected extracts, the type of extraction (solvent and polarity), the main bioactive compounds identified, and the final working concentrations based on the IC_50_ values obtained by the MTT assay.

In order to determine the potential of the extracts to alleviate the metabolic stress during obesity, we designed a panel of 45 genes related to metabolic processes including inflammation, oxidative stress, lipid, and cholesterol metabolism, glucose homeostasis, obesity, thermogenesis, and mitochondrial oxidative phosphorylation, to evaluate the effect of the extracts in the modulation of gene expression. We focus on genes, which have been shown to present genetic polymorphisms (SNPs) associated with the susceptibility to develop obesity and other metabolic processes with effects in weight, energy expenditure, thermogenesis, diabetes, secretion of pro-inflammatory cytokines, immune system function, or even nutrient-intolerances [37,38,39,40,41] with the final objective to evaluate the ability of the different extracts to modulate gene expression. Table 2 shows main metabolic pathways and genes associated with the different process, which were selected in the screening. The results of the screening are shown in Figure 2 (Heat map representation of the effect of the different extracts on gene expression).

As expected, the extracts affected in a different manner the expression of genes implicated in the metabolic pathways of interest (Figure 2). The successful application of Precision Nutrition requires the knowledge of the molecular mechanism of action of bioactive compounds in order to be proposed for specific subgroups of patients.

For this reason, we next strained our study on metabolic processes implicated in the activation of thermogenesis, and the enhancement of the mitochondrial respiratory capacity as complementary approaches to augment the energetic expenditure in obesity-related metabolic alterations [42,43]. Therefore, we selected 6 extracts, which included bioactive compounds representative of distinct families of polyphenols and flavonoids: Curcumin, Punicalagins from Pomegranate, Resveratrol from Grape, Flavonoids from Ginkgo Biloba, Isoflavones from Soy and Silymarin from Milk Thistle, to carry out in vitro functional studies. To evaluate the effects of the extracts on thermogenesis and/or mitochondrial oxidative phosphorylation, we selected two in vitro models: differentiated human adipocytes (SGBS human cell line) and differentiated human myocytes (HSMM human cell line), respectively.

### 3.2. Screening of Plant-Derived Extracts in the Expression of Genes Implicated in the Activation of Thermogenesis and Mitochondrial Oxidative Phosphorylation

#### 3.2.1. Effect of the Extracts on Cell Viability of Human Pre-Adipocytes and Human My-Blasts

In order to optimize the concentrations of the extracts to be assayed in differentiated adipocytes and myocytes, first, we performed MTT assays in non-differentiated proliferating cells to evaluate their effects on cell viability. As shown in Figure 3, four of the extracts had IC_50_ values higher than 100 mg/mL. On the contrary, curcumin and resveratrol extracts displayed IC_50_ values of 19.8 (+/− 4) mg/mL and 42.4+/− 4 mg/mL in proliferating pre-adipocytes, and 17.7 (+/− 1) mg/mL and 34.5 (+/− 3) mg/mL in proliferating myoblasts, respectively. To avoid deleterious effects due to the inhibition of cell viability, we continued only with 4 extracts, Thistle Silymarin, Pomegranate Punicalagins, Ginkgo Biloba, and Milk Soy Isoflavones, with IC_50_ values higher than 100 mg/mL. Thus, we studied the effects of the extracts in differentiated human adipocytes and differentiated human myocytes at two different doses (10 and 30 mg/mL) which did not compromise the cell viability.

#### 3.2.2. Effect of the Extracts in the Expression of Genes Related to the Activation of Thermogenesis and Mitochondrial Oxidative Phosphorylation

To evaluate the potential of the extracts in the activation of thermogenesis and mitochondrial oxidative capacity, first, we analyzed their effects in the modulation of the expression of the following genes: uncoupling proteins (*UCP1, UCP2, UCP3*) which are key players in the activation of mitochondrial thermogenesis [44,45,46,47,48,49,50,51,52]; PR domain containing 16 (*PRDM16*) and Bone morphogenetic protein 8B (*BMP8B*) [53,54] as browning activators; *PPARG* as the adipocyte differentiation inductor [55,56]; genes related to mitochondrial function and biogenesis, such as Peroxisome proliferator-activated receptor gamma coactivator 1-alpha (*PGC1A*) [57,58], Sirtuin 1 (*SIRT1*) [47,59,60,61], Mitochondrial Transcription Factor 1 (*TFAM1*) [62,63], 5’ adenosine monophosphate-activated protein kinase (*AMPK*) [30,64,65,66], and Mitochondrial creatine kinase (*CKMT*) [67,68]; genes implicated in glucose uptake and insulin homeostasis, such as Glucose transporter type 4 (*SLC2A4/GLUT4*) [69] and Insulin receptor substrate 1 (*IRS1*) [70,71]; genes related to lipid homeostasis [72] such as Sterol regulatory element-binding transcription factor 1 (*SREBF1*) [73,74], Fatty acid synthase (*FASN*) [75,76], fatty acid binding protein 4 (*FABP4*) [77,78], and Carnitine palmitoyltransferase I (*CPT1A*) [79,80]. We also included the analysis of genes related to the expression of well-known adipokines, batokines, and miokines, which are key mediators of lipid accumulation and/or inflammation at the systemic level [58,81,82], such as leptin (*LEP*) [83,84,85,86,87], Brain-derived neurotrophic factor (*BDNF*) and Fibronectin type III domain-containing protein 5 (*FNDC5*), which is the irisin precursor [88,89,90,91,92], and the pro-inflammatory Interleukin 6 (*IL6*), which contributes to promote lipolysis, and the uptake of nutrients in the muscle augmenting the energy expenditure while diminishing systemic inflammation [47,93,94].

Differentiated human adipocytes (SGBS) and myocytes (HSMM) were pre-treated with the extracts for 48 h at 10 and 30 mg/mL which were previously shown not to affect the cell viability (Figure 3).

Figure 4A illustrates genes modulated by the different extracts in differentiated SGBS human adipocytes, as well as a representation of the different cellular processes, mainly thermogenesis, browning and mitochondrial function and biogenesis, which may be influenced. Figure 4B shows only the statistically significant genes whose expression were affected after treatment differentiated SGBS human adipocytes with the extracts for 48 h.

Figure 5A illustrates genes modulated by the different extracts in differentiated HSMM human myocytes, as well as a representation of the different cellular processes, mainly thermogenesis, and mitochondria function and biogenesis, which may be influenced. Figure 5B shows only the statistically significant genes whose expression were affected after treatment differentiated HSMM human myocytes with the extracts for 48 h.

#### 3.2.3. Functional Analysis of Mitochondrial Oxidative Phosphorylation and Lipid Accumulation in Differentiated Adipocytes

As all the extracts modulated the expression of key metabolic genes implicated in the activation of thermogenesis and mitochondrial respiration, we next performed functional analysis by mean of the analysis of the extracellular oxygen consumption rate (OCR). In the MitoStress assay, after the sequential addition of modulators of the electron transport chain: oligomycin, FCCP, and Rotenone/Antimycin A, it is possible to determine key parameters of mitochondrial oxidative phosphorylation such as the basal OCR, the maximal OCR, the spare respiratory capacity, the ATP production linked to mitochondrial oxidative phosphorylation, and the H^+^ leak which is an indirect read out of the uncoupled respiration.

Being mitochondria key organelles controlling the cell bioenergetics, in this assay, we wanted to analyze the potential of the extracts to augment the mitochondrial oxidative capacity and the uncoupled respiration as mechanisms to increase the energy expenditure and/or to alleviate metabolic stress in obesity and related disorders [26,29,43].

Thus, differentiated human adipocytes were treated with the selected extracts Silymarin, Ginkgo, Punicalagin and Soy, for 48 h before conducting the MitoStress assay. In addition, it was included resveratrol as an internal control, since it has been reported to increase thermogenesis and mitochondrial function [95,96].

Figure 6A–D shows the mitochondrial respiration profiles of differentiated human adipocytes after treatment with the extracts. All the extracts augmented the spare respiratory capacity and the H^+^ leak compared to the control non-treated cells.

Interestingly, the extracts did not affect the basal OCR nor the maximal OCR, nor the H^+^ leak of non-differentiated human pre-adipocytes (Appendix A), suggesting the specific role of the extracts on the browning and thermogenic function of differentiated cells, without affecting the proliferative capacity of undifferentiated pre-adipocytes.

In accordance with these results, differentiated adipocytes pretreated with the extracts showed reduced neutral lipid accumulation (Red Oil Staining) and smaller lipid droplets compared to non-treated cells (Figure 6).

### 3.3. Functional Analysis of Mitochondrial Oxidative Phosphorylation in Differentiated Myocytes

To evaluate the effect of the extracts on the increased respiratory capacity of differentiated myocytes, as a mechanism to augment the energy expenditure, we pretreated differentiated HSMM cells for 48h at two different doses, before running the MitoStress assay.

Interestingly, the extracts did not affect the basal OCR nor the maximal OCR, nor the H^+^ leak of non-differentiated human myoblasts (Appendix A), suggesting the specific role of the extracts on the browning and thermogenic function of differentiated cells, without affecting the proliferative capacity of undifferentiated myoblasts.

Figure 7A–D shows the mitochondrial respiration profiles of differentiated human myocytes HSMM after treatment with the extracts. All the extracts augmented the basal OCR and spare respiratory capacity compared to the control non-treated cells.

## 4. Discussion

Nowadays, obesity is a pandemic that worldwide affects a large population, worsening multiple disorders such as cancer, CVD, metabolic syndrome, diabetes, and viral infections, among others [4,97]. Many of the comorbidities associated with obesity are related to the dysfunction of WAT, which promotes a low-grade of chronic inflammation [2,98]. In addition, although BAT depots are less susceptible to developing local inflammation, alterations in their functionality during obesity have important consequences in the systemic energetic balance [99].

For this reason, the search for new complementary strategies based on nutritional interventions to slow down the evolution of inflammatory processes and metabolic dysfunctions associated with obesity are being investigated [13,100,101,102]. Importantly, not all people have the same metabolic responses to specific nutritional interventions, as they have different genetic backgrounds. Therefore, it is crucial to develop personalized interventions taking into consideration the individual susceptibility to metabolic alterations and the molecular effects of bioactive compounds and nutrient-derived ingredients by means of the regulation of gene expression [15,103,104].

In this study, we have selected 20 extracts derived from natural sources (EFSA approved) from Natac Biotech (Table 1) to analyze their potential to modulate the expression of genes related to metabolic processes including inflammation, oxidative stress, lipid, and cholesterol metabolism, glucose homeostasis, obesity, thermogenesis, and mitochondrial oxidative phosphorylation. Specifically, we have focus on genes described to have genetic polymorphisms (SNPs) associated with the susceptibility to develop obesity and other metabolic processes with effects in weight, energy expenditure, thermogenesis, diabetes, secretion of pro-inflammatory cytokines, immune system function, or even nutrients intolerances [37,38,39,40,41], to evaluate the ability of natural extracts to modulate gene expression

As shown in Figure 2, the extracts affected in a different manner the expression of genes implicated in the metabolic pathways of interest. Indeed, for example, extracts from the same source, but with different main bioactive compounds, modulated gene expression in opposite directions. This is the case of anthocyanins and protocyanidins enriched extracts from grapes which downregulated the expression of *PGC1a, MTHFR, COL5A1, CLOCK, CD36, AHR, ABCA1, ADRB1, NPY, UCP2, MCM6, MC4R*. Meanwhile, resveratrol enriched extract from the same source upregulated their expression. Since individuals have distinct SNPs associated with the development of metabolic alterations and/or to the up- or downregulation of genes, distinct nutritional interventions will be required according to their genetics [101,104,105,106].

Next, we strained our study to investigate the potential of the extracts to tackle obesity and related metabolic alterations by augmenting the energy expenditure through the activation of thermogenesis and/or browning of WAT, and the enhancement of the mitochondrial respiratory capacity in muscles [42,43]. Therefore, we selected 6 extracts, including bioactive compounds representative of distinct families of polyphenols and flavonoids: *Punicalagins* from Pomegranate, *Flavonoids* from Ginkgo Biloba, *Isoflavones* from Soy, *Silymarin* from Milk Thistle, *Curcumin* and *Resveratrol* from Grape, to carry out in vitro functional studies. The analysis of the effects of the extracts on cell viability in non-differentiated proliferating adipocytes and myoblasts showed that curcumin and resveratrol had IC_50_ values below 45 mg/mL. To avoid deleterious effects, due to the inhibition of cell viability, we continued the analysis only with 4 extracts: *Milk Thistle Silymarin, Pomegranate Punicalagins, Ginkgo Biloba,* and *Soy Isoflavones* whose IC_50_ values were higher than 100 mg/mL (Figure 3).

Two of the most relevant tissues regulating the whole-body energy are adipose tissue and muscles. These two tissues control several processes related to inflammation, glucose homeostasis, lipid and cholesterol levels in plasma, insulin sensitivity and appetite, among others. Moreover, they are responsible, at least in part, for the regulation of thermogenesis in the body, a process which is also gaining importance as a therapeutic approach to control the metabolic and inflammatory stress in the course of obesity [26,107,108]. Hence, we analyzed the effects of the extracts in two in vitro models of differentiated human adipocytes (SGBS) and differentiated human myocytes (HSMM), in the modulation of the expression of metabolic genes related to thermogenic signaling pathways, lipid metabolism, nutrient oxidation, and mitochondrial function, among other features (Figure 4 and Figure 5) [44,46,53,55,59,63,74,78,80,81,82,84,109]. In addition, the functional relevance of the effects on gene expression was evaluated by means of the quantification of key parameters of the mitochondrial oxidative phosphorylation, including basal and spare respiratory capacities, H^+^ leak, and ATP production (Figure 6 and Figure 7A–D).

Milk Thistle extract enriched in Silymarin upregulated the expression of *UCP1, UCP2, CKMT2*, TFAM, and *PPARG* in differentiated adipocytes, which are implicated in the promotion of the browning process. Moreover, it upregulated the expression of *SIRT1* and *TFAM*, with a well-known role in the promotion of mitochondrial biogenesis (Figure 4).

The analysis of the mitochondrial function indicated a significant increase in the mitochondrial H^+^ leak on differentiated adipocytes (Figure 6A–D), which is a readout of thermogenesis. Importantly, lipid droplets were smaller, in line with the significant decrease in the expression of *SREBF1* and *GLUT4*, associated with *de novo* synthesis of fatty acids and the extracellular uptake of glucose, respectively. These results suggest a reduction of the metabolic stress associated with fatty acid accumulation as well as in the activation of the browning process in white adipocytes. Importantly, it was found a decrease in the expression of the pro-inflammatory cytokine IL6 as well as in the expression of LEP, indicating a rewire from a white adipocyte inflamed phenotype towards a browner functional one.

Silymarin extract also upregulated the expression of *UCP2*, *UCP3*, *SERCA1 PPARG, PGC1A, PRDM16*, *SIRT1,* and *TFAM* in differentiated myocytes (Figure 5), suggesting the stimulation of non-shivering thermogenesis and mitochondrial biogenesis.

In line with this, it augmented the basal and the spare respiratory capacity of differentiated myocytes, indicating increased functionality of differentiated myocytes (Figure 7). In summary, Silymarin promoted features of browning in human adipocytes, as well as an increase in the respiration capacity in differentiated myocytes, which together may contribute to increase the energy expenditure during obesity [2,98,110,111,112,113].

Regarding Ginkgo Biloba extract, gene expression levels of *UCP1*, *PGC1A,* and *BDNF* were upregulated in mature adipocytes, in line with the increased levels of H^+^ leak observed in mitochondrial function (Figure 6A–D). Furthermore, a significant decreased was found in the accumulation of neutral lipids (Figure 6E), as well as a decrease in the expression levels of the pro-inflammatory cytokine IL6 (Figure 4). In addition, this extract improved the performance of differentiated myocytes by means of the upregulation of the basal and spare respiratory capacities, in line with the increased expression of *CKMT2* and *SERCA1*. Similar results have also been reported in other extracts from Gingko Biloba [114,115].

Pomegranate extract enriched in punicalagins augmented the expression of *UCP1, UCP2, BMP8B,* and *CKMT2* in mature adipocytes (Figure 4), in line with the increase in the mitochondrial H^+^ leak (Figure 6A–D) and the reduced neutral lipid accumulation (Figure 6E), indicating a positive effect on the induction of thermogenesis and WAT browning. Similar results have also been suggested for other extracts from this source [116,117]. Although the expression levels of *CPT1a* were reduced, the reduced levels of neutral lipid accumulation suggest an alternative mechanism to enhance lipolysis. Differentiated myocytes treated with Pomegranate extract increased the expression levels of *PRDM16* (Figure 5), a key player in the reprogramming program from the myocyte precursor to the brown adipocyte [54,118]. Additionally, Pomegranate extract increased the expression levels of *BDNF,* a key myokine with autocrine and paracrine activities associated with the stimulation of lipolysis in AT while increasing muscle mitochondrial capacity [109]. Besides, this extract increased the expression of *IRS1* in differentiated myocytes, suggesting a role in the increase of muscle insulin sensitivity which is very important for glucose homeostasis. In addition, this extract improved the performance of differentiated myocytes by means of the upregulation of the basal and spare respiratory capacities.

Soy extracts rich in isoflavones significantly increased the expression levels of *UCP1, CIDEA,* and *CKMT2* (Figure 4), as well as the mitochondrial proton leak in differentiated human adipocytes in line with the reduced content of neutral lipid accumulation (Figure 6).

As genes related to the mitochondrial biogenesis such as *TFAM* were not affected, or even reduced such as *PGC1A*, it seems that this extract may have a role in the induction of thermogenesis in differentiated white adipocytes more than in the mitochondrial biogenesis associated with the browning process. This extract increased the expression levels of *IL6,* which could have some controversy regarding the inflammatory state, although it has been reported that at physiological levels, this cytokine may be a positive adipokine and myokine to induce lipolysis and thermogenesis in WAT. Although differentiated myocytes showed a reduced expression of *PGC1A* and *BDNF*, the increased mitochondrial respiratory capacity indicates additional metabolic pathways implicated, such as the improvement of insulin sensitivity [119,120,121,122].

One of the main limitations of this study is that the complete characterization of all the bioactive compounds present in the extracts has not been done. Nevertheless, in the last years, there is great interest in the study of bioactive extracts as mixtures of bioactive compounds targeting different metabolic pathways and exerting synergistic and higher bioactive effects. In addition, it remains to be explored the effect of the intestinal microbiome and the intestinal absorption and metabolization by intestinal and hepatic cells.

In conclusion, all the extracts could be useful in different scenarios depending on the individual characteristics. In this way, extracts that increase the expression of genes that promote biogenesis of mitochondria in the muscle, such as Milk Thistle and Pomegranate, may be of interest when developing strategies to improve the muscular aerobic capacity. Ginkgo and Pomegranate extracts may be relevant to reduce the inflammatory state in obese patients and in other metabolic alterations such as CVD and metabolic syndrome. Additionally, knowing that BAT presents higher basal oxygen consumption than WAT and a positive response to b-adrenergic stimulation [123], all the extracts seem to promote browning and thermogenesis in WAT and the development of a healthy beige tissue.

## Figures and Tables

**Figure 1 biomolecules-11-00412-f001:**
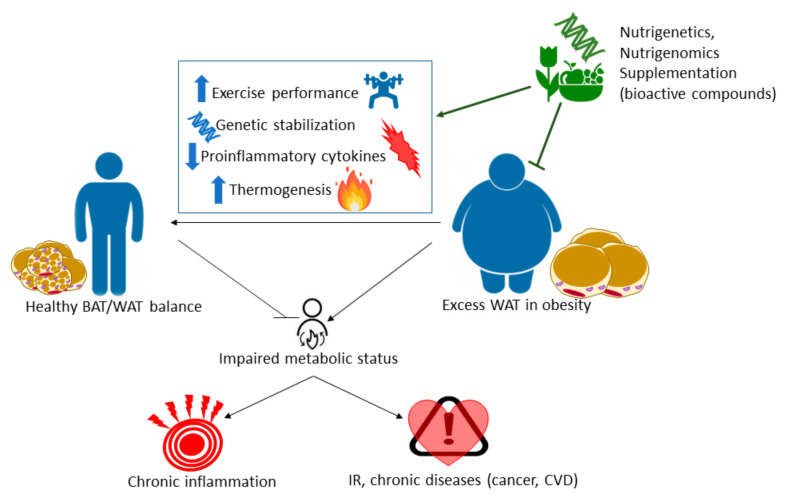
Precision Nutrition approach as a complementary approach in the treatment of metabolic diseases.

**Figure 2 biomolecules-11-00412-f002:**
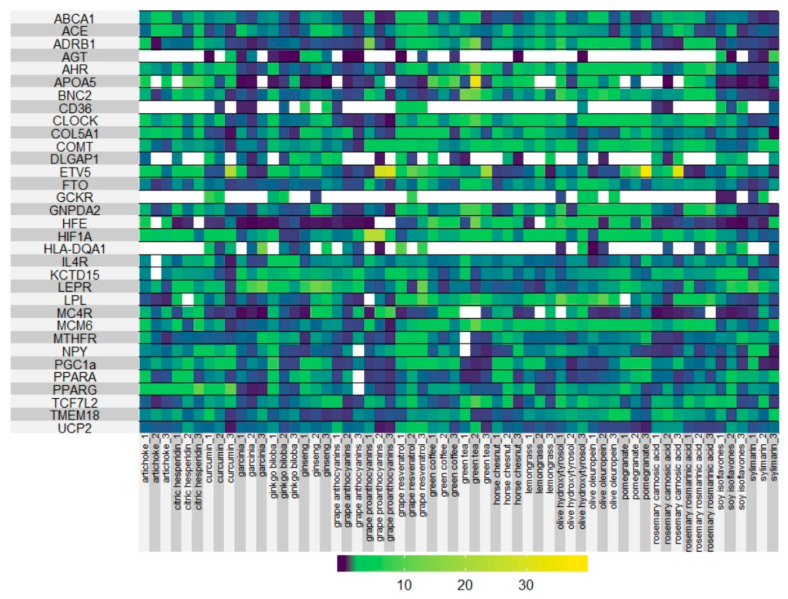
Heat map representation of the effects of the extracts (3 doses) on gene expression.

**Figure 3 biomolecules-11-00412-f003:**
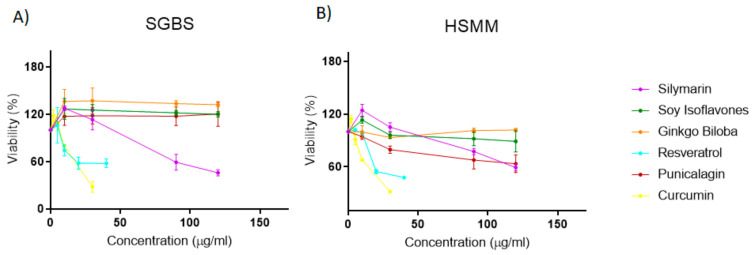
Effect of the extracts on cell viability (MTT assay) of (**A**) proliferating human pre-adipocytes (SGBS) and (**B**) proliferating human myoblasts (HSMM).

**Figure 4 biomolecules-11-00412-f004:**
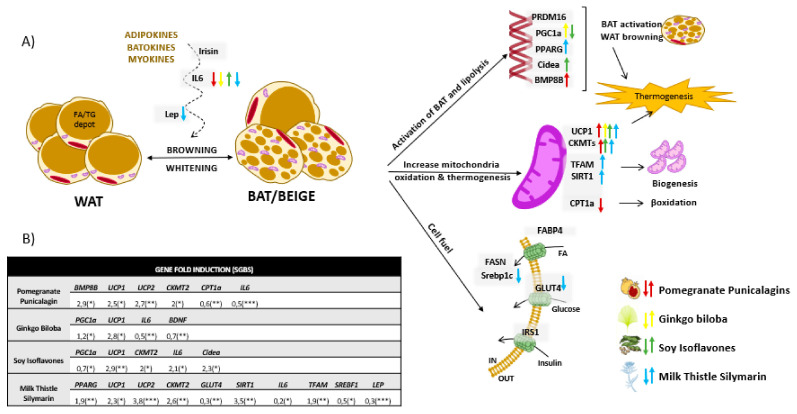
(**A**) Graphical representation of genes modulated by the extracts in differentiated human adipocytes SGBS compared to no-treated cells. Up or down arrows indicate upregulation or dow-regulation in gene expression after treatment with the extracts. Statistically significant genes whose expression was modulated in differentiated human adipocytes SGBS after 48h of treatment with the extracts, compared to no-treated cells. (**B**) Mean fold induction of at least 3 independent experiments are shown. Asterisks indicate significant statistic differences * < 0.01, ** < 0.005, *** < 0.001.

**Figure 5 biomolecules-11-00412-f005:**
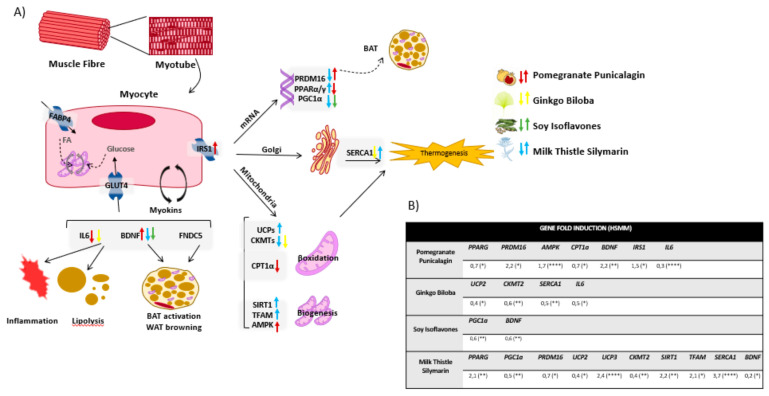
(**A**) Graphical representation of genes modulated by the extracts in differentiated human myocytes HSMM compared to no-treated cells. Up or down arrows indicate upregulation or downregulation in gene expression after treatment with the extracts. Statistically significant genes whose expression was modulated in differentiated myocytes HSMM after 48h of treatment with the extracts, compared to non-treated cells. (**B**) Mean fold induction of at least 3 independent experiments are shown. Asterisks indicate significant statistic differences * < 0.01, ** < 0.005, **** < 0.0001.

**Figure 6 biomolecules-11-00412-f006:**
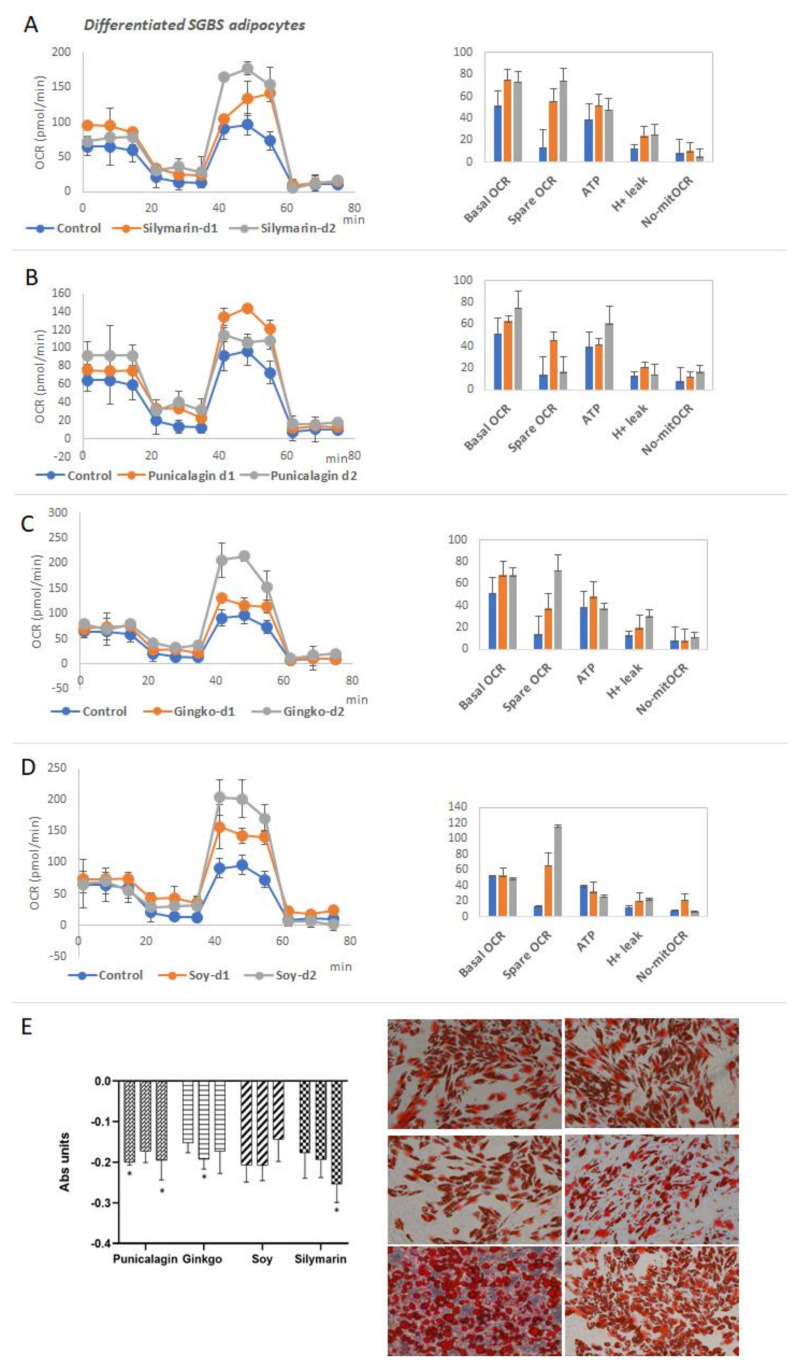
Effect of the extracts on mitochondrial oxidative phosphorylation of differentiated human adipocytes. (**A**–**D**) MitoStress profiles of the Oxygen Consumption Rate (OCR) after the sequential addition of modulators of the electron transport chain of differentiated adipocytes treated with different extracts ((**A**) treated with Silymarin extract, (**B**) Punicalagin, (**C**) Ginkgo, and (**D**) Soy, respectively). Quantification of basal OCR, Spare respiratory capacity, ATP production linked to oxidative phosphorylation, H^+^ leak and non-mitochondrial OCR. Asterisks indicate significant statistic differences * < 0.01. (**E**) Red Oil O staining and quantification after treatment with the extracts. Representative images of Red Oil staining to show lipid droplets.

**Figure 7 biomolecules-11-00412-f007:**
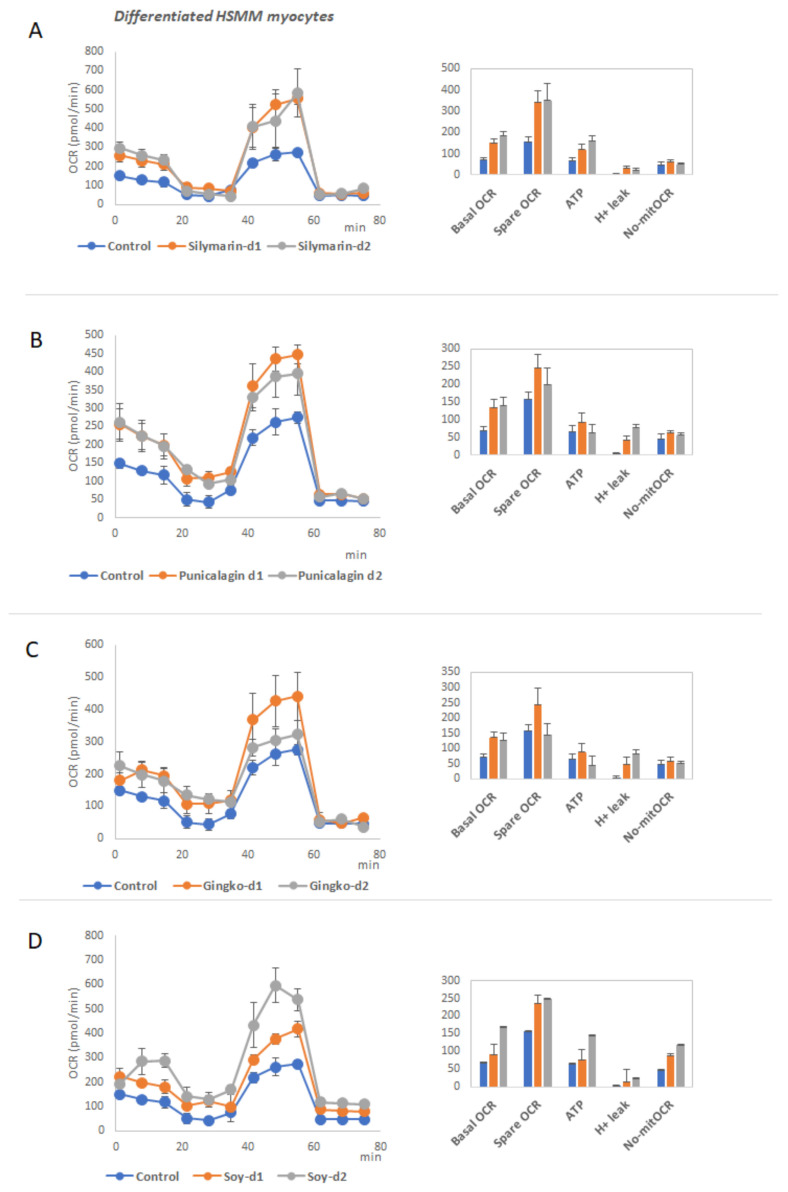
Effect of the extracts on mitochondrial oxidative phosphorylation of differentiated human myocytes. (**A**–**D**) MitoStress profile of the OCR after the sequential addition of modulators of the electron transport chain of differentiated myocytes treated with different extracts (Figure 7A treated with Silymarin extract, Figure 7B Punicalagin, Figure 7C Ginkgo, and Figure 7D Soy, respectively). Quantification of basal OCR, Spare respiratory capacity, ATP production linked to oxidative phosphorylation, H^+^ leak and non-mitochondrial OCR.

**Table 1 biomolecules-11-00412-t001:** List of the 20 selected extracts indicating their natural source, their main bioactive compounds, the parts of the plant used for the extraction, the extraction solvents, and the final working concentrations used based on the IC_50_ values obtained by the MTT assay.

*Terpenes*	*Saponins*	*Aesculus hippocastanum* L. (Horse chesnut)	Seed	Ethanol: water (70:30 V/V)	Aescin	10, 30, 90	>100
		*Panax ginseng* C.A.Mey (Ginseng)	Root	Ethanol: water (50:50 V/V)	Ginsenoside	10, 30, 90	>100
	*Diterpenes*	*Rosmarinus officinalis* L. (Rosemary)	Leaves	Ethanol	Carnosic acid	5, 10, 15	≈18
*Organic acids*		*Garcinia* *Cambogia* L. (Garcinia)	Fruit	Water	Hydroxycitric acid	10, 30, 90	>100
*Polyphenols*	*Phenolic acids*	*Cynara scolymus* L. (Artichoke)	Leaves	Ethanol: water (60:40 V/V)	Hydroxycinnamic derivatives	10, 30, 90	>100
		*Coffea Arabica* L. (Green coffee)	Fruit	Ethanol: water (60:40 V/V)	Hydroxycinnamic derivatives	10, 30, 90	>100
		*Punica granatum* L. (Pomegranate)	Fruit	Ethanol: water (70:30 V/V)	Punicalagin	10, 30, 90	>100
		*Olea europaea* L. (Olive)	Fruit	Water	Hydroxytyrosol	10, 30, 50	≈60
		*Olea europaea* L. (Olive)	Leaves	Ethanol: water (70:30 V/V)	Oleuropein	10, 30, 90	>100
		*Rosmarinus officinalis* L. (Rosemary)	Leaves	Ethanol: water (50:50 V/V)	Rosmarinic acid	10, 30, 90	>100
	*Silbenes*	*Aloysia citrodora* L. (Lemon verbena)	Leaves	Ethanol: water (60:40 V/V)	Verbascoside	10, 30, 90	>100
		*Vitis Vinifera* L. (Grape)	Root	Ethanol: water (50:50 V/V)	Resveratrol	5, 10, 20	≈20
	*Diarylheptanoids*	*Curcuma longa* L. (Tumeric)	Root	Ethanol	Curcuminoids	2, 5, 10	≈8
	*Flavonoids*	*Ginkgo biloba* L. (Ginkgo)	Leaves	Ethanol: water (70:30 V/V)	Flavonoid glycosides	10, 30, 90	>100
		*Glycine max.* (L.) Merr (Soy)	Seed	Ethanol: water (50:50 V/V)	Isoflavone	10, 30, 90	>100
		*Vitis Vinifera* L. (Grape)	Fruit	Water	Anthocyanins	10, 30, 90	>100
		*Camellia sinensis* L. (Green tea)	Leaves	Ethanol: water (60:40 V/V)	Flavan-3-ols	10, 30, 90	>100
		*Vitis Vinifera* L. (Grape)	Seed	Water	Proanthocyanins	10, 30, 70	≈75
		*Citrus* sp. (Orange)	Fruit	Ethanol: water (50:50 V/V)	Flavonoid glycosides (Hesperidin)	10, 30, 90	>100
		*Silybum marianum* L. Gaertn. (Milk thistle)	Seed	Ethanol: water (70:30 V/V)	Silymarin	10, 30, 90	>100

**Table 2 biomolecules-11-00412-t002:** List of the genes included in the customed TaqMan gene expression customized array cards (TLDAs) cards analysis, grouped in metabolic signaling pathways related to cell bioenergetics, glucose metabolism, lipid metabolism, inflammation, and metabolic stress, among others.

Metabolic Process	Genes Implicated
Energetic metabolism, thermogenesis, and obesity	*UCP2, HIF1A, PGC1a, FTO, MC4R, KCTD15, ETV5, NPY,*
Xenobiotic metabolism	*AHR*
Lipid metabolism and adipogenesis	*LEPR, LPL, PPARG, PPARA, PGC1a, AHR, LIPC, FTO, CD36*
Cholesterol and lipoprotein homeostasis	*LPL, PPARA, ABCA1, APOA5*
Glucose homeostasis and diabetes	*TCF7L2, PGC1a, TMEM18, GNPDA2, GCKR*
Blood pressure	*AGT, ACE*
Mitochondrial biogenesis	*PPARG, PGC1a*
Metabolic response to exercise and muscle capacity	*MTHFR, PPARG, HFE, HIF1A, PGC1a*
Aerobic capacity and sport performance	*ADRB1, ACE, HIF1A*
Structural function	*COL5A1,*
Biorhythm	*CLOCK*
Appetite	*NPY, LEPR*
Immune response	*IL4R, PPARA*
Senescence and aging-related diseases	*MTHFR, ABCA1, BNC2, COL51A, DLGAP1, CD36,*
Cognitive function	*DLGAP1, COMT, ETV5, NPY*
Food-related intolerances (gluten, lactose, fructose)	*HLA-DQA1, MCM6*

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
