# Peer review of "Natural Extracts to Augment Energy Expenditure as a Complementary Approach to Tackle Obesity and Associated Metabolic Alterations"

_biomolecules, 2021, doi:10.3390/biom11030412_

Round 1

Reviewer 1 Report

The manuscript has improved considerably; the authors have finalized their paper following the indications suggested by the reviewers

I am very satisfied with the corrections and additions made by the authors.

In my opinion, the manuscript can now be published.

Author Response

Thank you for all the comments which have allowed to improve the Manuscript.

Reviewer 2 Report

Authors performed the suggested corerctions. The manuscript is now improved. 

Author Response

(The authors gave the same response as above.)

Reviewer 3 Report

Manuscript Biomolecules -1139376 titled “Natural extracts to augment energy expenditure as a complementary approach to tackle obesity and associated metabolic alterations” by Marina Reguero et al., is a research article and its aim is to investigate molecular targets of bioactive natural extracts to ameliorate issues associated to obesity. The extracts used by the authors were selected from the library of Natac Biotech.

The paper investigates an interesting topic. The text is clear and quite easy to read. The results are clearly presented, although Supplementary Figures 1 and 2 are not visible. The methods are adequately described. The Authors used appropriate statistic methods.  The conclusions are consistent with presented arguments. References are up to date.

However, some points have to be overcome.

  • Authors should explain better the need to use of HEK293 cells.
  • Section: 2.5. Gene expression analysis. Last line, it is not table “2”.
  • Authors should merge tables 1 and 2 (delete one)
  • Figure 6. Authors should add letters a, b and c in the figures and add more information in the caption.
  • Authors should review the references, modifying it according to the instructions of the Biomolecules journal

Authors should check the color of the words in the text (some words are in red, others in blue)

Author Response

Please, see attached document.

This manuscript is a resubmission of an earlier submission. The following is a list of the peer review reports and author responses from that submission.

Round 1

Reviewer 1 Report

Although the manuscript by Reguero and colleagues addresses an important topic, many critical issues need to be clarified.

In fact, the authors discuss the effects of some "natural" molecules on some cellular modeling. In particular, the authors evaluate the expression of some genes involved in multiple and heterogeneous pathophysiological mechanisms.

Some critical issues need to be clarified by the authors.

1) Authors should redo table 1. Indeed, authors must use the official nomenclature to define the plant species used. Furthermore, it is not clear how the concentrations of the different molecules were chosen. What is the source of the chosen molecules? What type of extraction was done? Were commercial standards used? This is a methodological passage that should be clarified by the authors.

2) The entire manuscript is based on the effects of extracts of plant origin but it is not clear the type of extracts used. The authors should know that the type of solvents used for the extraction can affect the type of metabolite present in the sample.

3) I suggest some references that could help you in your evaluations:

Gupta AK et al., A. Artocarpus lakoocha Roxb. and Artocarpus heterophyllus Lam. Flowers: New Sources of Bioactive Compounds. Plants (Basel). 2020

Mastinu A et al., . Gamma-oryzanol Prevents LPS-induced Brain Inflammation and Cognitive Impairment in Adult Mice. Nutrients. 2019

Reviewer 2 Report

In this article, I did not find which species or a certain biologically active molecule is effective for obesity.

For the special role of Biomolecules, I think it is necessary to explore specific active molecules instead of evaluating extracts in general.

So I recommend submitting to other journal (Nutrition or Foods of MDPI)

Reviewer 3 Report

General comment: Research article entitled “Natural extracts to augment energy expenditure as a complementary approach to tackle obesity and associated metabolic alterations” presents the possible role of specific natural extracts on obesity by using in vitro models with adipocytes. This is an interesting study, with sufficient methodology. Some major corrections are required for the improvement of the manuscript.

Abstract: The Abstract adequately presents the background and the aim of the review article.

-Authors should reconsider the structure in order to contain less theoretical information and more data about the study design and the basic results.

Introduction: The introduction section is well-written and covers the importance to further investigate new approaches for the therapy obesity.

-Authors could state more clearly the aim of the study in the last paragraph.

Material and methods: The material and methods are adequately presented.

-Authors should describe in a separate session the natural extra used (more info).

Results and discussion:  Authors adequately and analytically present and discuss the results of the study.

-Table 1 could be located at Methodology session.

-Could authors shortly discuss possible limitations of the study?

References: The references used by the authors cover adequately the relative scientific field and the aims of the study.